# Identification and initial response to children's exposure to intimate partner violence: a qualitative synthesis of the perspectives of children, mothers and professionals

Natalia V Lewis,[1,2] Gene S Feder,[1] Emma Howarth,[3] Eszter Szilassy,[1] Jill R McTavish,[4] Harriet L MacMillan,[4,5] Nadine Wathen[6]

For numbered affiliations see end of article.

**Correspondence to**
Dr Natalia V Lewis;
nat.lewis@bristol.ac.uk

## ABSTRACT

**Objectives** To synthesise evidence on the acceptable identification and initial response to children's exposure to intimate partner violence (IPV) from the perspectives of providers and recipients of healthcare and social services.

**Design** We conducted a thematic synthesis of qualitative research, appraised the included studies with the modified Critical Appraisal Skills Programme checklist and undertook a sensitivity analysis of the studies scored above 15.

**Data sources** We searched eight electronic databases, checked references and citations and contacted authors of the included studies.

**Eligibility criteria** We included qualitative studies with children, parents and providers of healthcare or social services about their experiences of identification or initial responses to children's exposure to IPV. Papers that have not been peer-reviewed were excluded as well as non-English papers.

**Results** Searches identified 2039 records; 11 studies met inclusion criteria. Integrated perspectives of 42 children, 212 mothers and 251 professionals showed that sufficient training and support for professionals, good patient-professional relationship and supportive environment for patient/clients need to be in place before enquiry/disclosure of children's exposure to IPV should occur. Providers and recipients of care favour a phased enquiry about IPV initiated by healthcare professionals, which focuses on 'safety at home' and is integrated into the context of the consultation or visit. Participants agreed that an acceptable initial response prioritises child safety and includes emotional support, education about IPV and signposting to IPV services. Participants had conflicting perspectives on what constitutes acceptable engagement with children and management of safety. Sensitivity analysis produced similar results.

**Conclusions** Healthcare and social service professionals should receive sufficient training and ongoing individual and system-level support to provide acceptable identification of and initial response to children's exposure to IPV. Ideal identification and responses should use a phased approach to enquiry and the WHO Listen, Inquire about needs and concerns, Validate, Enhance safety and

### Strengths and limitations of this study

► This is the first synthesis of qualitative studies focusing on the integrated perspectives of patients/clients and healthcare and social service professionals on the acceptable identification and initial response to children's exposure to intimate partner violence.

► We retrieved relevant studies through a comprehensive search strategy, including electronic searches, citation and reference checking and contacting experts.

► Involvement of two reviewers throughout screening, data abstraction and critical appraisal of each study ensured methodological rigour of this review. Reviewers' backgrounds in different disciplines broadened and enriched the interpretation of data.

► Thematic synthesis allowed us to: (i) integrate perspectives of all participant groups and generate new interpretations going beyond the findings from primary studies, (ii) identify gaps in evidence within and across participant groups; (iii) establish areas of conflicting perspectives, which can be targeted in future research and interventions.

► Exclusion of non-English papers and non-peer-reviewed reports could result in missing some relevant studies; methodological limitations of the included studies weakened the reliability and objectivity of the evidence.

Support principles integrated into a trauma-informed and violence-informed model of care.

## INTRODUCTION

Intimate partner violence (IPV) is a violation of human rights and widespread public health problem that is associated with impairment throughout the lifespan. It is defined as any behaviour by a current or former intimate partner associated with physical, sexual or psychological harm, including

acts of physical aggression, sexual coercion, psychological abuse and controlling behaviours.[1] Although IPV is experienced by both men and women, the morbidity and mortality related to IPV is highest among women.[2] IPV has detrimental effects on physical, mental and reproductive health of women and has a negative impact on their children.[3 4] Children's exposure to IPV can occur through direct involvement and witnessing or through indirect exposure (eg, being aware of the violence between parents/caregivers, financial consequences, parenting affected by IPV).[5 6] In the US, the prevalence of child witnessing a parent assaulting another parent is 5.8% in the past year and 25% over the life-time.[7] Children's exposure to IPV is strongly associated with a broad range of emotional and behavioural problems, including internalising and externalising symptoms, as well as increased risk-taking behaviour and academic problems. Furthermore, such exposure among children can lead to physical health consequences, including injuries and death, when physical violence between caregivers directly involves children.[8 9] Healthcare professionals (HCPs) and social service professionals (SSPs) have an important role in identifying and responding to adult patients/clients and their children exposed to IPV.[10–12]

Identification of children's exposure to IPV in healthcare or social service settings can occur when the abused parent or caregiver seeks help, when children undergo assessment for behavioural problems or when other services notify healthcare and social service providers about IPV occurrence. The WHO IPV guidelines recommend a case-finding approach to identify women exposed to IPV: healthcare providers asking those women who present with indicators or clinical associations of IPV about safety in their relationship and at home.[1] There is no equivalent guidance on effective and acceptable approaches to identifying and responding to children's exposure to IPV and limited evidence on which to base that guidance.[8]

This review is one of a series undertaken for the Violence, Evidence, Guidance and Action (VEGA) Project, informing pan-Canadian public health guidance on family violence.[13] The objectives of the present systematic review were to identify, appraise and synthesise research evidence on the acceptability of the identification and initial responses to children's exposure to IPV in healthcare and social service settings. The synthesis addressed the following research questions:

1. What approaches to identification of children's exposure to IPV are acceptable to children, non-abusing parents and professionals?
2. What initial responses to children identified as being exposed to IPV are acceptable to children, non-abusing parents and professionals?

## METHODS

We focused on qualitative evidence because we wanted to understand how providers and recipients of care perceive approaches to identification and responses to children's exposure to IPV and why they find them acceptable or otherwise.[14] Qualitative research explores peoples' own experiences and perspectives through analysing textual or visual material obtained while talking to people or observing them.[15] This systematic review follows the Centre for Reviews and Dissemination[16] and Preferred Reporting Items for Systematic Reviews and Meta-Analyses[17] guidance and adheres to the ENhancing Transparency in REporting the synthesis of Qualitative research (ENTREQ) checklist.[18]

### Search methods

We aimed to retrieve relevant studies in the field through a comprehensive search and sampling strategy,[19] building on an earlier review by Howarth *et al* in the IMPRoving Outcomes for children exposed to domestic ViolencE (IMPROVE) 2013 evidence synthesis.[20] First, we retrieved full-text reports assessed in the IMPROVE synthesis. Second, we re-ran IMPROVE searches (28 April 2016) in eight medical, social science, social care and nursing databases (Ovid MEDLINE, PsychINFO, The Cochrane Library, Embase, Web of Science Social Sciences Citation Index, Web of Science Conference Proceedings Citation Index- Social Science & Humanities, Social care online, CINAHL on EBSCO) (see online supplementary file 1). Finally, the first reviewer completed forward and backward citation chaining of all included papers and emailed corresponding authors of the included papers (12, 24 August 2016) asking to confirm peer-reviewed status of reports and signpost to additional relevant papers.

### Studies selection

Inclusion criteria are summarised in table 1.

Multiple papers from the same study were included if they each reported new data relevant to the research questions. Exclusion of non-English papers and papers that have not been through the formal peer-review system (eg, books, conference papers, editorials, letters, general comment papers) was justified by limited resources and concerns about validity and reliability of non-peer-reviewed sources, respectively.

Two reviewers independently screened titles and abstracts of all references. The first reviewer screened all full-text papers, the second reviewers screened a 10% subset and disagreements (27%) were resolved through discussion and consensus.

### Analysis

#### Quality appraisal

Two reviewers independently assessed each study for methodological validity using the Critical Appraisal Skills Programme Qualitative checklist[21] modified for the purpose of the VEGA Project (M-CASP, score range 0–20).[22] Disagreements not resolvable between reviewers were resolved by a third reviewer. We did not exclude studies on reporting quality[23]; we conducted a secondary sensitivity analysis by restricting the synthesis to studies

**Table 1** Inclusion criteria for selecting studies

| Category | Inclusion criteria |
|---|---|
| Population (further called stakeholders) | ▶ Providers of healthcare (healthcare professionals (HCPs)) or social services (social service professionals (SSPs)) OR<br>▶ Recipients of healthcare or social services (further called patients or clients, respectively):<br>– Children (however defined in primary studies) who have been exposed to intimate partner violence (IPV) OR<br>– Non-abusing parents of children who have been exposed to IPV |
| Intervention | ▶ Identification of children' exposure to IPV by any method(s)—screening, case-finding, notification by other services, self-disclosure OR<br>▶ Initial response to children's exposure to IPV that followed the identification and occurred before referral to another professional or service |
| Phenomena of interest | ▶ Views of and direct experiences with identification and initial response to children's exposure to IPV |
| Types of studies | ▶ Publication date: database inception to 28 April 2016 AND<br>▶ English language AND<br>▶ Empirical qualitative (standalone or components of mixed-methods research) AND<br>▶ Qualitative methods for data collection and analysis (eg, interviews, focus groups, observations) AND<br>▶ Verbatim quotations from participants AND<br>▶ Papers that have undergone formal peer-review |

SSPs cover a range of services provided to advance adult and child welfare including child protection services.

in the top tertile of methodological quality (M-CASP score≥15).[24]

### Data extraction

We adapted a data extraction form from IMPROVE.[20] The first reviewer extracted study details; second reviewers checked the extracts. The papers were then treated as primary text transcripts. Where studies included varied participants, only data relevant to our inclusion criteria were considered. Two reviewers independently extracted raw qualitative data[25] relevant to the views and direct experience of identification and initial response to children's exposure to IPV from 'Results' section of the included papers. These data could be in the form of participants' quotes or authors' interpretations of participants' voices. For each study, the reviewer entered data extracts into the form separately for children, parents and professionals.

### Synthesis

Our choice of synthesis method was guided by the practical aims of this review, the 'thickness' of the data reported in the primary studies and the expertise of the team.[26 27] We applied thematic synthesis[25 28] in three stages:

I. *Line-by-line coding* alongside data extraction. Two reviewers independently coded the data extracts for themes relevant to the acceptability of the identification approach and initial response to children's exposure to IPV[29] subsequently meeting to compare and combine their codes. The first reviewer produced a final table of codes with supporting verbatim text for each participant group in each study.

II. *Developing descriptive themes.* The first reviewer grouped the codes into themes and subthemes with accompanying verbatim to capture consistency and range of views within each participant group and across the studies.[29] Second reviewers commented on the table leading to the final version.

III. *Generating analytical themes.* The first reviewer used the constant comparison method[30] to integrate perspectives across child, parent and professional groups. The integrated stakeholder perspectives were categorised by level of agreement within and across the groups.[31] When perspectives on a theme were consistent, it was categorised as convergent. When stakeholders' views on a theme were consistent within groups, but differed between them, it was categorised as divergent. Finally, themes with wide variation, within and between the groups, were categorised as conflicting. This integration through categorisation produced interim analytical themes which were further refined in relation to the research questions. Our approach allowed us to: (i) integrate perspectives of all participant groups and generate new interpretations beyond findings from primary studies, (ii) identify gaps in evidence within and across participant groups and (ii) establish areas of conflicting perspectives, which can be targeted in future research and interventions. Throughout this stage, the first reviewer developed diagrams and tables with interim analytical themes, which were refined during group discussions at three meetings with all second reviewers.

### Patient and public involvement

Patients and public were not involved in the design and conduct of this secondary analysis of published research.

## RESULTS

Sixteen papers reporting 11 studies were included (figure 1); three were reported in multiple papers. Two papers were from a study on parents' experiences with Irish child protection services.[32][33] Two papers drew on a study of police and children's social services responses to IPV incidents where children were present or resided in the household.[34][35] Four papers were from the Researching Education to Strengthen Primary care on Domestic Violence and Safeguarding (RESPONDS) study on general practice clinicians' perspectives on child safeguarding in IPV cases.[36–39] All papers were published between 2008 and 2015.

### Studies characteristics and methodological quality

The 11 included studies are summarised in table 2; online supplementary file 2 contains detailed studies characteristics. Ten studies were based in high-income countries: the UK,[35][39][40] the USA,[41–43] Australia,[44][45] Ireland[32] and Canada.[46] One study was based in a middle-income country, Brazil.[47]

Voices of children were reported in two studies,[35][45] parents in eight studies[32][35][40][42–46] and professionals in seven studies.[35][39–42][46][47] Overall, the studies involved 42 children and young people aged 8–24 years (19 from IPV and social services, 23 from general practice), 220 parents (212 mothers) and 251 professionals (113 healthcare, 42 social services and 96 mixed samples). All parents were IPV survivors. HCPs included physicians[37][41] and nurses[37][42][47] from primary and secondary healthcare. SSPs were drawn from children's social services,[35] child protection services[46] and unspecified settings.[40][41]

Of 11 studies, 7 studies[30][39–44] scored ≥15 out of 20 on the M-CASP indicating their overall good methodological quality (figure 2, online supplementary file 3). The main shortcomings identified were in the M-CASP domains of reliability and objectivity. Thus, the authors did not justify their choices of study design,[35][37][40][41][44][47] research methods,[35][37][40–43][46][47] participant selection[35][40][41][43][44][46][47] and recruitment.[33][35][37][40][42][43][47] Only two studies[41][45] described strategies for establishing neutrality.

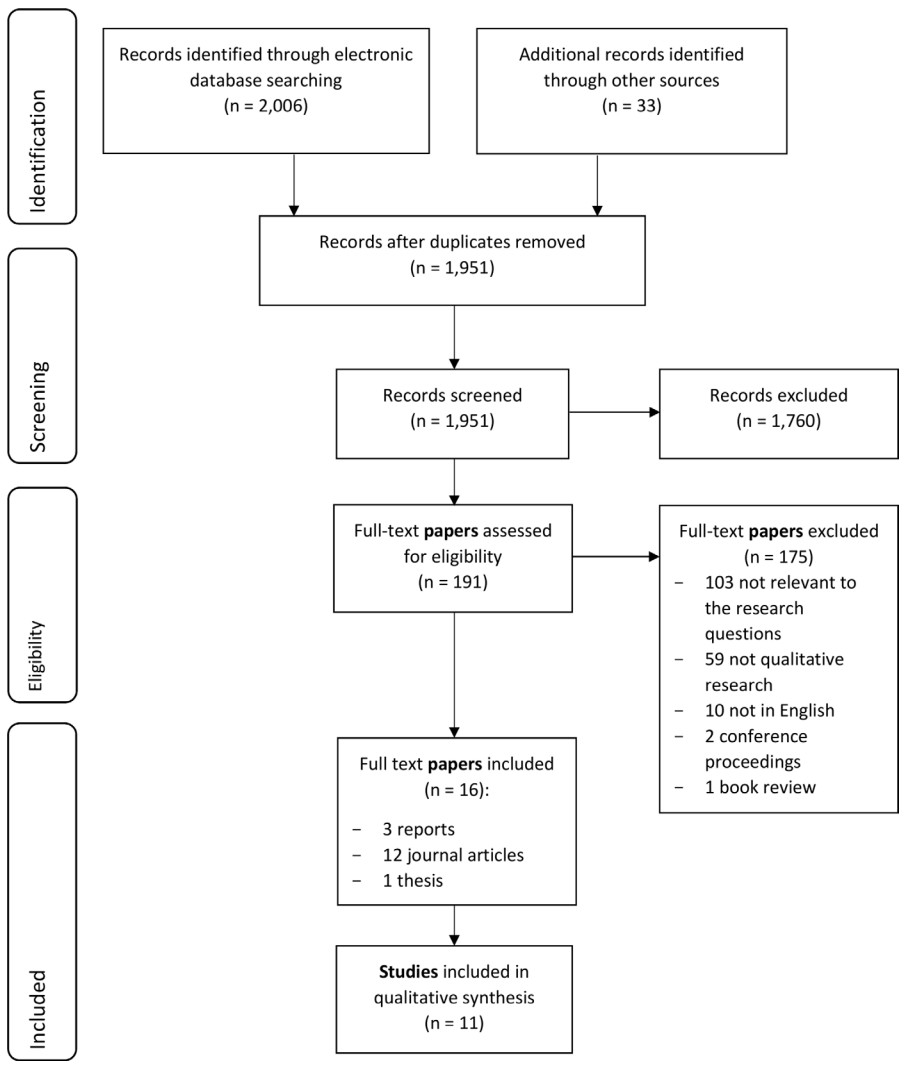

**Figure 1** Flow of studies through the review.

**Table 2** Summary characteristics of the 11 studies (16 papers) synthesised (in chronological order)

| Study | Country | Topic | Methodology | Children | Parents | HCPs | SSPs |
|---|---|---|---|---|---|---|---|
| | | | | **Stakeholder group** | | | |
| Buckley et al[32 33] | Ireland | Experiences of child protection services | Interviews | | X | | |
| Black et al[41] | USA | Interventions for IPV | Interviews | | | X | X |
| Stanley et al[34 35] | UK | Police IPV notifications of children's social services | Interviews | X | X | | X |
| Meyer[44] | Australia | Help-seeking of IPV victims with children | Interviews | | X | | |
| Randell et al[43] | USA | IPV information in healthcare setting | Focus groups | | X | | |
| Davidov et al[42] | USA | Mandatory reporting of children's exposure to IPV | Secondary analysis of interviews and focus groups | X | X | X | |
| Angelo et al[47] | Brazil | Experiences of providing care to children exposed to IPV | Interviews | | | X | |
| Jenney et al[46] | Canada | Communication between providers and recipients of child protection service | Interviews and focus groups | X | | | X |
| Szilassy et al[36–39] | UK | Experiences of responding to children's exposure to IPV | Interviews | | | X | |
| Clarke and Wydall[40] | UK | Experiences of responding to children's exposure to IPV | Interviews, focus groups, observations | X | X | | X |
| Morris[45] | Australia | Safety and resilience of children exposed to IPV | Interviews and focus groups | X | X | | |

IPV, intimate partner violence; HCPs, healthcare professionals; SSPs, social service professionals.

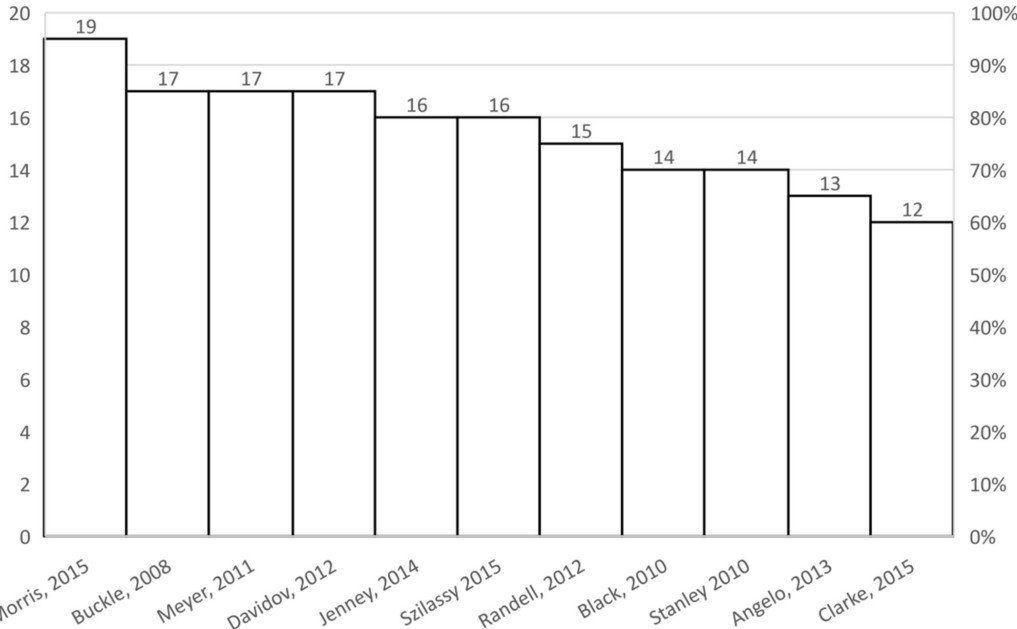

**Figure 2** Methodological quality of the studies as assessed by the modified Critical Appraisal Skills Programme (score range 0–20). Studies scored ≥15 are in the top tertile.

## Synthesis

### Line-by-line coding and descriptive themes

Initial line-by-line coding generated 75 codes. We grouped them into 22 descriptive themes with 13 subthemes related to: (1) experiences of identifying children's exposure to IPV; (2) experiences of the initial response to children's exposure; (3) factors enabling identification and initial response to children's exposure; (4) reasons for not identifying or disclosing children's exposure or not engaging with services; (5) psychological consequences of individuals' involvement in identification and initial response to children's exposure and (6) suggested training and resources (see online supplementary file 4). All relevant parents' quotations were from mothers who had experienced IPV.

### Analytical themes

Categorisation of participant views by agreement/disagreement within and across the three stakeholder groups showed converging, diverging and conflicting perspectives on satisfactory identification and responses to children's exposure to IPV. These perspectives were expressed in five interim analytical themes (see online supplementary file 5) that were refined further, when possible resolving conflicting perspectives within and across the studies. Finally, we articulated the final analytical themes as descriptions of an ideal identification and initial response:

1. Precursors for acceptable identification and response;
2. Acceptable identification;
3. Acceptable initial response;
4. Conflicting perspectives on engagement with children and management of safety (table 3).

Not all participant groups contributed equally to the final analytical themes. Thus, professionals' perspectives were presented across 15/17 subthemes, while mothers informed 13/17 and children 9/17. Children's quotes were not available for most subthemes covering an acceptable initial response. Quotes from each stakeholder group supporting the final analytical themes are collated in the online supplementary file 6.

### *Precursors for acceptable identification and response*

This theme was developed from converging perspectives on facilitators and barriers to satisfactory identification and response. It captures areas of agreement within and between stakeholder groups on the enabling processes and conditions required before children's exposure to IPV is disclosed or discussed. Mothers and children know and trust professionals who are non-judgemental and non-threatening, have good communication skills, can ensure confidentiality and can offer practical help.[32 35 37 41 42 44–46] Good communication and trusting patient-professional relationship make mothers and children feel comfortable and safe in discussing sensitive issues and to return for support, if needed.[32 45] Professionals match their approaches to the individual mother's readiness to disclose and engage with services and work with the mother towards increasing her readiness.[35 41 43 44 46] Culturally sensitive materials in different languages on IPV and children's exposure to IPV are displayed in healthcare settings to prepare patients for enquiry and provide information for those who are not ready to disclose and engage.[41 43]

Professionals receive sufficient training and guidance built on local policies on children's exposure to IPV and interagency work with children's social services and IPV services.[37 41 42] Training fits into daily practice of HCPs and SSPs and prepares them to better communicate with children.[37] Training and guidance clarify: (i) what constitutes children's exposure to IPV, especially psychological and non-direct physical, (ii) what are professionals' roles and reporting duties, (iii) how and where to document children's exposure to IPV and (iv) how and where to signpost mothers and children.[32 35–37 41–46] Training and guidance increase professional awareness, skills and confidence, which lead to more satisfactory identification and responses.

Mothers, children and professionals receive emotional support and help with managing the emotional burden of involvement in identifying and responding to IPV. For patients/clients, such support is provided through non-judgemental responses and confidentiality (or being clear when reports must be made).[32 34 35 43 45] For professionals, ongoing emotional support and supervision need to be arranged.[35 37 41 47] This helps to maintain well-being and mental health of patients and professionals and leads to increased satisfaction.

The acceptability of any work related to identification of and response to children's exposure to IPV can be undermined by systems-level constraints, such as high demand on services, lack of resources and system support and poor interagency collaboration.[35 37 40 41] Therefore, the approaches to identification and responses must be aligned with the capabilities of under-resourced healthcare and social services.

### *Acceptable identification*

This theme was developed from converging perspectives on the acceptable approaches to identification. It captures areas of agreement within and between stakeholders on how professionals should ask about children's exposure to IPV. The enquiry is built on a trusting patient-professional relationship, takes place in a safe and supportive environment and is integrated into the context of the consultation or visit.[37 41 42 45 47] Acceptable identification involves a trained and well-supported HCP who is non-judgemental and has good communication skills.[32 35 37 41 42 44–46] HCP uses a phased approach, that is, starts from the presenting symptoms, then moves to general safety and well-being of the child and finally asks about feeling safe at home.[37 41 42 45 47]

### *Acceptable initial responses*

This theme was developed from converging perspectives on what the acceptable initial response should include

**Table 3** Final analytical themes and their definitions

| Final analytical themes with subthemes | Definition | Stakeholder group, study | | |
|---|---|---|---|---|
| | | Children | Mothers | Professionals |
| **1. Precursors for acceptable identification and response** | | | | |
| 1.1. Satisfying and sustainable relationship | Patients know and trust professionals with whom they develop good long-term relationships. Trusting relationships enable patients to feel safe and comfortable to discuss sensitive issues. | 45 | 32 44–46 | 37 41 46 |
| 1.2. Desired professional attitudes and skills | When interacting with patients/clients, professionals demonstrate non-judgemental, non-threatening attitudes, show respect, actively listen, validate patient's accounts, reassure confidentiality and provide practical help. | 35 45 | 32 35 42 44–46 | 37 41 42 |
| 1.3. Considering mother's readiness | Professionals acknowledge individual mothers' readiness to disclose IPV and engage with services, work towards increasing mothers' readiness and match their approaches to the stage of mothers' readiness. | | 43 44 | 35 41 46 |
| 1.4. Patient materials | Culturally sensitive materials on IPV and children's exposure to IPV in different languages are displayed in healthcare settings. | | 43 | 41 |
| 1.5. Professional training | Professionals receive adequate training on communication with children, indicators of children's exposure to IPV, especially psychological and non-direct physical IPV, professionals' role in identifying and responding, documenting and reporting, interagency work. | 35 45 | 32 35 43–46 | 35–37 41 42 |
| 1.6. Professional resources | Professionals have clear guidance on local IPV resources, what constitutes children's exposure to IPV, what is reportable and how to document children's exposure to IPV in a way that keeps the child safe and ensures the safety and confidentiality of the mother. | | | 37 41 42 |
| 1.7. Professional supervision and support | Professionals have skilled supervision and ongoing support for coping with psychological consequences of working with children and mothers exposed to IPV and preventing vicarious trauma | | | 35 37 41 47 |
| 1.8. Addressing systems' barriers | Professionals' work of identifying and responding to children's exposure to IPV fits into the organisational, local and national context of increased demands on healthcare and social services without commensurate resources. | | | 35 37 40 41 |
| **2. Acceptable identification** | | | | |
| 2.1. Space and time | It is ideal to give patients permission, space and time to discuss sensitive matters. | 45 | 45 | |
| 2.2. Vocabulary | It is preferable for HCPs to phrase questions about children's exposure to IPV as a 'safety-at-home' matter. | 45 | 45 | |
| 2.2. Phased approach | When asking about children's exposure to IPV, it is ideal for HCPs to initiate the enquiry, adapt it to the context of the consultation and use a phased approach—from presenting symptoms to general safety and well-being, then to safety at home. | 45 | 45 | 37 41 42 47 |
| **3. Acceptable initial response** | | | | |
| 3.1. Shifting focus | Professionals first focus their responses on the mother-child dyad and shift to the child if he/she is at risk of harm. Professionals need assistance with managing emotional burdens caused by the shift. | | | 35 37 41 |
| 3.2. Emotional support | When responding to disclosure, it is ideal to provide children and parents with encouragement and emotional support. | | 46 | 37 39 47 |

**Table 3** Continued

| Final analytical themes with subthemes | Definition | Stakeholder group, study | | |
|---|---|---|---|---|
| | | Children | Mothers | Professionals |
| 3.3. Education | It is acceptable to educate mothers about the impact of IPV on children, IPV dynamics, professionals' roles and duties in responding. However, education should not jeopardise patient safety (eg, through sending materials home where the perpetrator can find them). | | 43 | 35 37 41 42 46 |
| 3.4. Signposting | It is acceptable for professionals to give children and mothers information about local IPV services. | 35 | 35 | 35 37 46 |
| 4. Conflicting perspectives on engagement with children and management of safety | | | | |
| 4.1. Engaging directly with children | Stakeholders' perspectives on the acceptability of talking directly to children exposed to IPV and seeing them alone are conflicting. Children are absent in the patient-professional communication. Mothers and children want direct engagement with children. Professionals do not see children as patients on their own and feel ill-equipped for communicating with children about IPV. | 34 35 45 | 45 | 35 37 40 |
| 4.2. Management of safety | Stakeholder preferences regarding risk assessment and safety planning are conflicting. Mothers and children are absent in the management of safety and want to be involved. Professionals are not satisfied with current risk assessment and safety planning approaches and want them to change. | 45 | 46 | 35 37 40 41 |

IPV, intimate partner violence; HCPs, healthcare professionals; SSPs, social care professionals.

and how it should be delivered. It also captures conflicting professional perspectives on the focus of their responses, reported within and between three studies[35 37 41] and resolved though justifying that child safety is prioritised over mother's safety and confidentiality. Due to the overlap between IPV and children's exposure to IPV, most HCPs first target their initial response to the mother-child dyad.[37 41] However, their focus shifts towards the child if they are at risk of harm.[35 37] Some professionals feel conflicted when prioritising the child's needs over the adult's and require assistance with managing the emotional burden.[41] Professionals educate the mother about the impact of IPV on children, as well as the roles and duties of different providers.[35 37 41–43 46] This helps to relieve mothers' fears and increase their readiness to engage with services. Importantly, the approach to mothers' education should not jeopardise their safety (eg, by sending materials home).[35] Mothers and children must also be given information about local IPV services.[35 37 46]

*Conflicting perspectives on engagement with children and management of safety*

This theme captures divergent and conflicting perspectives on the appropriateness of engaging directly with children[35 37 40 45] and satisfactory management of safety.[35 37 40 45 46] The variations were reported within and between studies and remain unresolved in our review. While most children and mothers are positive about professionals talking directly to children and addressing their individual needs, a few held the opposite view.[35 45] In contrast, most professionals do not feel competent in

communicating directly with children and prefer to assess children's needs through a proxy adult,[37 39] while some report always engaging with children and some think they should engage more.[35 40] Furthermore, all stakeholder groups have different views on the appropriate age for talking directly to children about IPV.[37 39 45] For example, some mothers think that HCPs can talk to their children about IPV from the age of 5, while some doubt the acceptability of such conversations even with adolescents.[45] Finally, children and HCPs do not agree on the appropriate age for seeing a doctor alone. Most children prefer their mother or trusted adult to be with them at the consultation and indicate that at least teenagers should be seen alone.[45] Similarly, most HCPs indicate they would not see children alone, with some suggesting that if they were to do so, they would need to be adolescents. A few HCPs suggest that they would seek a parent's and child's permission to see the child alone, but they do not elaborate on the practicalities of obtaining such permission.[37]

Next, mothers and SSPs have divergent and conflicting views on the management of child safety and both perceive current practices as unsatisfactory. First, safety is understood differently by mothers and professionals. Actions perceived by mothers as increasing child's safety (eg, staying with perpetrator for financial reasons, not seeking help to prevent escalation of abuse) are seen by SSPs as increasing the risk for the child.[44 46] While most mothers think that involvement of children's social services increases the risk for the child through potential escalation of abuse and child removal, most SSPs believe

that their involvement results in greater safety. Whereas most SSPs think that women and children are safer out of the abusive relationship and feel frustrated when women do not follow their advice to leave, women do not feel safer after leaving the perpetrator because of potential escalation of abuse, child contact with perpetrator and lack of postseparation support. However, mothers feel intimidated by SSPs and that they must adhere to their instructions to keep children in the home and minimise involvement of children's social services.[32 44 46]

Second, stakeholders' perspectives on acceptable risk assessment and safety planning vary and diverge. Children's preferences for the assessment of risk in all environments are represented by one boy's voice, which does not match any of the professional or parent viewpoints.[45] Mothers report not being included in safety planning with SSPs and want their views to be considered.[46] Professionals have polarised perspectives on the acceptability of the current risk assessment and safety planning with most feeling dissatisfied with the inconsistent approaches and bureaucracy involved.[35 37 40 41 46] A few SSPs suggest engaging with mothers and children more without elaborating on how this should be done.[40] All stakeholders report feeling conflicted about the existing management of safety, but have to comply due to the imbalance of power and system requirements.[35 37 40 41 46]

### Sensitivity analysis

Exclusion of four studies scoring <15 on the M-CASP[35 40 41 47] did not change the final analytical themes.

### DISCUSSION
### Main findings

This synthesis included 11 qualitative studies with 513 providers and recipients of healthcare and social services. We identified enabling precursors, ideal approaches and areas of disagreement among children, mothers and professionals regarding the acceptable identification and initial response to children's exposure to IPV. Enabling precursor processes were linked to patient/client-professional relationship building, creating a safe and supporting environment and changing and matching responses according to individual mothers' readiness to disclose and engage with services. Enabling conditions included sufficient training and multiple language-versions materials and embedding the work of identifying and responding to children's exposure to IPV into the context of under-resourced services. Acceptable identification involved a phased approach to enquiry. An ideal initial response included emotional support, education about IPV and signposting to IPV services. Areas of disagreement were related to the acceptability of engaging directly with children and managing child's safety.

An important finding of this synthesis is that participants' views on many factors related to satisfactory identification and initial response to children's exposure to IPV are strikingly consistent. Our first theme suggests

that mothers and children need to have a trusting relationship with a professional who demonstrates certain attitudes and skills before enquiry and identification occurs. Combined with the theme of a phased approach to enquiry, our findings support a case-finding approach to identifying IPV. Our third theme supports the acceptability of providing mothers and children with emotional support, education about IPV and signposting to local IPV services. These essential components of good clinical practice on identification and response to IPV are supported in the literature[2 24 48] and highlight the importance of safe environments and trusting relationships as conditions for recognition of and response to mothers' and children's exposure to IPV.[49] Furthermore, the acceptability of a case-finding approach is consistent with the approach outlined by WHO on identification of child maltreatment in the recent mhGAP Intervention Guide.[50] The ideal initial responses to children's exposure to IPV are in line with the WHO Listen, Inquire about needs and concerns, Validate, Enhance safety and Support (LIVES) principles for IPV response.[1]

Another notable finding is the role of the wider context. Although most of the included studies were based in high-income countries, all professionals described how their ability to identify and respond to children's exposure to IPV was heavily influenced by healthcare and social service systems constraints. Specifically, they felt they lacked time to engage with their patients about sensitive issues; they felt burdened by constant cuts and restructuring of healthcare and social services and they expressed frustration about poor access to referral pathways. These systemic factors were compounded by poor communication and coordination across organisations and absence of a single referral pathway, consistent with findings in research on child protection services.[51 52] This highlights the importance of targeting interventions on identification and initial response to children's exposure to IPV at both individual and system levels,[53–55] with professional guidance adaptable to the changing landscape of local services.

Our findings about the emotional burden of IPV work suggest that identifying and responding to children's exposure to IPV can have negative psychological impact on both providers and recipients of care (eg, disempowerment, compassion fatigue and vicarious trauma). The common causes of distress are patients'/clients' feelings of shame and guilt linked to the acknowledgement of IPV and disclosure, professionals' ambiguous feelings towards mothers who did not follow their advice, tensions when shifting the focus from mother-child dyad to the child and frustration with system-level obstacles. These findings, which are consistent with previous research,[56–58] emphasise the importance of assisting both patients/ clients and professionals with managing psychological symptoms and preventing vicarious trauma.[58–60] The results also supports a trauma-informed and violence-informed approach to care of adults and children exposed to violence.[59–62]

One of the most striking findings of this review is the gap between including children's own needs and preferences, and the lack of attention paid by professionals to children as patient/clients on their own. Although professionals recognised the importance of working with a mother-child dyad exposed to IPV, most did not perceive children as individual patients/clients and used a proxy adult to assess and respond to the child's needs. In contrast, mothers and children were in favour of HCPs talking directly to children, treating each one as an individual and addressing the child's needs directly. The invisibility of children's own views has been previously reported[63–67] and can be explained by ethical and methodological challenges of undertaking research with children,[68] rigid professional attitudes and lack of knowledge/competence on communicating with children. However, the finding is still concerning, especially given growing recognition of children's rights[69] and agency.[70] The gap between patient and professional preferences for talking directly to children suggests that professionals need training and guidance on communicating with children about sensitive issues. With regard to children's exposure to IPV specifically, professionals need help understanding safety requirements, such as asking children in a private, confidential environment about exposure to IPV.[50]

Patients/clients and professionals held conflicting views about the acceptable management of safety. Neither patients nor professionals were satisfied with current risk assessment and safety planning. Addressing the needs of mother and child involves aligning professionals' and mothers' diverging perspectives about safety and risk. While professionals may think that leaving an abusive partner is a prerequisite for safety, this is the time when women are at greatest risk of homicide[71]; furthermore, women themselves know when it is safest to leave.[72] In our synthesis, professionals favoured targeting the mother-child dyad, but found it acceptable to switch the focus to the child when they are at risk of harm. This is consistent with literature, which suggests a hierarchy of needs for children and caregivers, recognising the needs of both mother and child, but prioritising the child given children's inherent vulnerability.[73]

While not the focus of this synthesis, it is important to note that mandatory reporting of child maltreatment laws may complicate or intertwine with strategies for inquiring about exposure to IPV, as children's exposure to IPV is a reportable exposure in some jurisdictions.[74 75] We found that most HCPs were confused as to whether children's exposure to IPV was reportable in their jurisdiction. They felt anxiety about the reporting duties and thresholds. This finding is in line with the recent meta-synthesis on mandated reporters' experiences with reporting child maltreatment.[22] The authors offer recommendations for mitigating potential harms associated with the reporting processes. The strategies include disclosing reporting duties and the limits of confidentiality when providers start a relationship, consulting with child protection services in an anonymous manner when a provider is unsure if the suspected maltreatment is reportable, and—if the suspected maltreatment is reportable—discussing with the child/family how the provider will file a report (when it is safe for the child to do so) and likely child protection service responses to the report. Discussing reporting duties and limits of confidentiality should ideally occur before inquiry, to minimise feelings of betrayal that may emerge when a provider realises they must report. These recommended strategies are applicable to the reportable children's exposure to IPV.

### Strengths and limitations

We used comprehensive strategies for retrieving papers, including systematic searches of bibliographic databases, citation searching, reference checking and emailing topic experts. Involvement of two reviewers throughout screening, data abstraction and critical appraisal reduced potential bias. Reviewers' backgrounds in different disciplines broadened and enriched data interpretation. Bringing together perspectives of children, mothers and professionals gave the recipients of care a voice equal to the providers. Our analytical themes are easily understood and can be used by practitioners and policy makers as targets for interventions on identification and response to children's exposure to IPV.

This review has several limitations. Only papers published in English were included due to limited resources. This limitation alongside the exclusion of books and conference abstracts could result in missing studies relevant to the review questions. However, our decision to focus on papers that have been peer-reviewed and potentially of better methodological quality, increases the robustness of our findings.

The evidence we produced should be interpreted with caution, taking into consideration the following limitations of the primary data. It is supported by only 11 studies from high-income and middle-income countries, 4 of which had methodological shortcomings regarding reliability and 9 were lacking objectivity. Some important information (eg, mandatory reporting status of professionals) was not specified. The selective nature of the stakeholders' sample makes our findings relevant to HCPs and children's social service providers, mothers who have experienced IPV and their children. All evidence supported by children's voices came from two studies. Although small, the sample of children covered varied ages and settings.

Future studies should explore children's own experiences of encounters with healthcare and social services around identification and initial responses to IPV. Children's voices will provide important information on their values and preferences. Future studies should also recruit fathers who have experienced IPV. Our fourth theme of conflicting perspectives should guide future research on the acceptable approaches to talking directly to children about IPV and undertaking risk assessment and safety planning.

## CONCLUSION

The present analysis adds to the evidence base important, client-centred considerations drawn from qualitative research to enhance approaches to the identification of, and response to, children exposed to IPV, and their caregivers. Healthcare and social service professionals should receive sufficient training and ongoing individual and system-level support to recognise and respond safely to children's exposure to IPV. Ideal identification and response should be based on the WHO LIVES principles,[12] integrated into a trauma-informed and violence-informed model of care.[76] In Canada, findings from the present analysis have been integrated into professional guidance and curriculum in the VEGA Project.[13]

**Author affiliations**
[1]Centre for Academic Primary Care, Population Health Sciences, Bristol Medical School, University of Bristol, Bristol, UK
[2]Centre for Primary Care and Public Health, Blizard Institute, Barts and the London School of Medicine and Dentistry, Queen Mary University of London, London, UK
[3]NIHR Collaboration for Leadership in Applied Health Research & Care (CLAHRC) East of England, Cambridge Institute of Public Health, University of Cambridge, Cambridge, UK
[4]Department of Psychiatry and Behavioural Neurosciences, McMaster University, Hamilton, Ontario, Canada
[5]Department of Pediatrics, McMaster University, Hamilton, Ontario, Canada
[6]Faculty of Information and Media Studies, and Centre for Research & Education on Violence Against Women & Children, Western University, London, Ontario, Canada

**Acknowledgements** The authors would like to thank Dr Lisa Arai for performing the electronic searches and for helping with title and abstract screening. The authors are grateful to Khabo Piggott and Eleanor Wright who volunteered to help with data extraction, coding and critical appraisal.

**Contributors** HLM and NW conceived the idea. GSF, NVL, EH, ES designed the study. NVL identified additional studies, carried out title and abstract screening, full-text screening, quality appraisal, data extraction, coding and synthesis. She reported the research and drafted the paper. EH, ES, GSF conducted double full text screening, data extraction, coding and quality appraisal. NVL, EH, ES, GSF, JRM contributed to the development of descriptive and analytical themes. All authors contributed to the interpretation of the data and five revisions of the paper.

**Funding** This work was supported by the Public Health Agency of Canada, through funding to the VEGA Project (HLM, NW). HLM was supported by the Chedoke Health Chair in Child Psychiatry. NVL was also supported by the Avon Primary Care Research Collaborative and the National Institute for Health Research (NIHR) Collaboration for Leadership in Applied Health Research and Care North Thames at Bart's Health NHS Trust (NIHR CLAHRC North Thames).

**Disclaimer** The views expressed in this article are those of the author(s) and not necessarily those of the National Health Service, the National Institute for Health Research or the Department of Health and Social Care.

**Competing interests** None declared.

**Patient consent** Not required.

**Provenance and peer review** Not commissioned; externally peer reviewed.

**Data sharing statement** Data extracted from each of the 16 papers are stored in Word files and are available from the corresponding author on reasonable request. Raw syntheses of these data are stored in Excel and Word files and are available from the corresponding author on reasonable request.

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
