## [Reviewer comments · BMJ Open]

ARTICLE DETAILS

TITLE (PROVISIONAL)	IDENTIFICATION AND INITIAL RESPONSE TO CHILDREN'S EXPOSURE TO INTIMATE PARTNER VIOLENCE: A QUALITATIVE SYNTHESIS OF THE PERSPECTIVES OF CHILDREN, MOTHERS AND PROFESSIONALS
AUTHORS	Lewis, Natalia; Feder, Gene; Howarth, Emma; Szilassy, Eszter; McTavish, Jill; MacMillan, Harriet; Wathen, Nadine

VERSION 1 – REVIEW

REVIEWER	Tuija Helena Leppäkoski, PhD University of Tampere, Finland
REVIEW RETURNED	12-Oct-2017

GENERAL COMMENTS	The prevention of all kinds of child maltreatment, including IPV, is a part of universal family services and child protection. The Authors have done a great job. I found some shortcomings and I suggest correcting them before publishing. The following are detailed comments. Introduction: Theoretical background is a little bit parsimonious. First, the concept and content of IPV (also called DV) was not clearly and accurately defined. Second, current research data supports the need to consider men and women as both potential victims and perpetrators when approaching IPV (e.g. Int J Public Health 2015;60:467-478; J Fam Viol 30:699-717). This study emphasize that the women are victims and the men are perpetrators. The choice of this point of view is necessary to justify. Second, violence against the child within the family is a serious and complex phenomenon and there are many factors that influence it. Further, it is often concealed. For this reason, I suggest that the authors will address the types and consequences of violence in a little more detail. (E.g. WHO; INSPIRE. (/www.who.int/violence_injury_prevention/violence/inspire/en/) Further, IPV effects on the health and well-being of the whole family. Methods: The methods used are transparent. This systematic review follows CRG guidance and it is ok. However, the structure of Methods section is somewhat confusing. Sometimes, it is difficult to understand and evaluate the points (paragraphs) the authors have made. It contains a lot of sub-headings and the text breaks down. Is it possible to combine them into larger entities? For example: search methods, selection criteria, data extraction and synthesis. Timeframe in which the literature was selected is not mentioned in the text. Moreover, there are many supplementary files, which must be read
--

simultaneously with the text to understand what the authors have done. I think that supplementary file 3 (list of excluded papers and reasons for exclusion) is completely unnecessary and exclusion criteria can be said in the text. "Studies reported in books ... were excluded as well as languages other than English, quantitative research and conference proceedings" (Page 5).
Supplementary file 5 (Characteristics of the 11 studies ...) there could be added two columns that briefly summarize the key findings of the each study and the M-CASP score.

Results:

Things that appear in the first two sub-chapters (Searches & Critical appraisal) are perhaps better suited to the Methods section. For example, the flow chart of study selection process (Fig 1) as well as study quality assessment could be placed under Methods section. In the first chapter there is something contradictory between the text and source references (four papers are mentioned but three sources 32-34, lines 15-17) and between the text and supplementary file (Supplementary file 3 comprises list of excluded papers and reasons for exclusion. The text that begins with the words "Two papers were from ..." can be placed well under Study characteristics.

Otherwise, the results are quite clearly presented.

Discussion:

What can be said about the generalizability and transferability of the research results? It is important to tell readers something about these.

On page 17 it is said that review has several limitations but only two (only papers published in English are included & only two of eleven studies reported children's voices) are mentioned. For example, the women survivors are over-represented in the review material.

Further, "All relevant parents' quotations were from mothers", (page 9). This also might be a gap.

Already known, that both men and women involved in IPV as both victims and perpetrators. Also mothers abuse their children and spouses, but this is not always revealed. Professional's obligation is to identify fathers' symptoms and signs of IPV as well, and provide services for them. In the future, it is important to extend research in this direction. This topic as a future research area, have not discussed.

One big problem is a late identification of abused children. Thus it is important to ask all family members how the everyday life goes, especially, when different life's problems intertwine. This is necessary to do in all the units that deal with children's issues. For this there are internationally tested, valid risk assessment tools (forms) to facilitate discussions with parents around this sensitive issue (e.g. CAP; Child Abuse Potential). On the other hand, why not ask directly if you have any doubt about IPV? Silence is not helping children and their parents. Naturally, children's rights, personal experience and involvement in care around IPV services are important research areas as the Authors have stated.

Other remarks:

- Page 24. The title of Figure 1 is missing. Likewise, there is no source reference. Where does this flowchart come from? Page 25. The figure number is missing.
- The reference to Table 2 is missing from the text page 9.
- In Supplementary file 5, Characteristics of 11 studies (page 46), the reference number is obviously wrong. Szilassy (2015) is number [39] not [34].
- Otherwise the figures and tables support the text.

	What this paper adds? What is new understanding and / or knowledge did this paper bring in compared to the previous ones concerning identification of and responding to children exposed to IPV, especially in international level? This was somewhat unclear and requires clarification. I hope that my comments will help the Authors to improve the article before it comes to publishing.
--	--

REVIEWER	John Devaney School of Social Sciences, Education and Social Work Queen's University Belfast Northern Ireland BT7 1NN
REVIEW RETURNED	16-Oct-2017

GENERAL COMMENTS	This is an interesting and timely paper, seeking to present the current best evidence on issues relating to the identification and initial response to children exposed to intimate partner violence. The paper is well written and the level of attention to the study design and data extraction is appropriate. I have some minor queries; 1) p6 A 10% subset of the full text searches were checked - what level of agreement was there between the first and second reviewer? 2) In some jurisdictions, such as the USA, child protection workers are not social workers. As such using the term 'social services' or 'social care practitioners' might appear misleading, as they are typically referred to as 'child protection services' and 'child protection workers'. This needs better explained in the opening sections as to how 'health practitioners' and 'social care practitioners' are defined. 3) Given that the paper deals with the issue of identification and initial response there is a need to at least acknowledge that some of the included studies took place in countries with mandatory laws for the reporting of child maltreatment, which might include children's exposure to domestic violence. This may influence whether professionals enquire about certain issues, and the required response. This needs discussed, as the context of the included studies may influence and moderate the findings. 4) Some of the papers (eg Stanley) are about the criminal justice response a
---

REVIEWER	Emiko Tajima, Associate Professor University of Washington, USA
REVIEW RETURNED	19-Oct-2017

GENERAL COMMENTS	This manuscript is a meta synthesis of 11 qualitative studies on the topic of identifying and responding to Intimate Partner Violence (IPV) exposure in children in health and social service settings. The authors have sifted through a large number of studies (over 2000 records and 191 full text papers) to identify 11 qualitative studies (described in 16 papers). The authors have been thorough in addressing the requisite elements of a rigorous meta synthesis, including clear search terms, sources, and methods, clear criteria for inclusion, clear analytic approach and systematic assessment of study rigor. The synthesis is comprehensive and thematic findings are presented in an easily digestible manner, being shown in both
---

	table outline form and in more detailed, narrative form. This thoroughness has yielded a comprehensive, yet lengthy manuscript with some repetitiveness. The authors focused on two areas: “acceptable approaches” to identifying IPV exposure in children, and “initial responses” to identified IPV exposure in different settings, including clinical settings, social service departments, IPV shelters, and child welfare settings. They sought to examine experiences and perspectives of service providers, victims of IPV, and children. While it is commendable to examine experiences in a range of settings such as in the current synthesis, it also complicates the interpretation and application of the findings because the settings are quite different, and the context is therefore different. For example, the identification of IPV exposure in the context of a child protective services office is different from identification within a domestic violence shelter, and different still from identification in a doctor’s office. The context differs, as do the potential consequences of identification. Professional service providers across those different settings have different perspectives, training, and understandings of IPV, and have very different roles in relation to the caregivers and their children. What works best and is perceived as most appropriate in one setting may not be regarded as appropriate or desirable in another setting. It is also not clear the level of “crisis” that the respondents might have been in at the time they participated in the studies. This aspect of the context matters as well. To their credit, the authors made an effort to examine cultural context factors. Even greater attention to race and culture in such a synthesis would be a welcome addition to the literature. It is not clear to me whether the practitioners who participated in the studies were all mandated reporters of child abuse – it complicates the synthesis that it’s unclear whether the precursors identified apply to mandated reporters of child abuse or to any practitioner in the studied settings. It is notable that practitioners reported considerable confusion themselves about whether or not they needed to report identified child abuse, and lack of clarity about how to define reportable child abuse, including whether or not IPV exposure itself was considered child abuse. I would not define IPV exposure as child abuse per se, however it is clear from the quotations that at least one of the practitioners defined it as such. This synthesis offers critical findings about the extent of confusion among practitioners about reporting requirements and documentation recommendations – this is something that administrators and policy makers should address. I would also note that for practitioners who suspect child abuse or neglect and who are mandated reporters, while they may seek to build rapport with the caregiver and child, and while it may be ideal to be able to reach a joint decision to contact child protective services (and even make the call together with the IPV victim caregiver), this may not be possible or feasible. It is a strength of the synthesis that the authors sought to explore the perspectives of multiple stakeholders, namely health and social service providers, caregivers who experienced IPV, and their children. It is useful to better understand patient / client and constituent perspectives and use this knowledge to inform clinical practice and policy. It is particularly valuable to see the points of alignment and also the areas of disagreement, especially comparing professionals and clients. Although one would hope that hospital and social service administrators would be interested in patient/client perspectives and preferences, they may be more responsive to data
--	---

	on positive outcomes than to data on patient/client beliefs and preferences. To change policy and practice, it may be necessary to also synthesize findings on factors that increase positive outcomes that administrators and policy-makers may care most about (e.g., positive patient health outcomes, reduced IPV, greater awareness of IPV effects on children, fewer hospital visits etc.). This meta synthesis is well implemented and makes an important contribution to the literature on the practice implications of children's exposure to IPV. The manuscript offers valuable findings, synthesized across 11 studies that reveal important practice and policy recommendations. Importantly, the synthesis across multiple studies underscores key systemic factors such as scarce resources and the limited time currently afforded to health and social service providers, all of which shapes practice and exacerbates the challenge of effective identification and response to IPV in these services settings.
--	---

REVIEWER	Jordan Sieboni a. Service Universitaire de Psychiatrie de l'Adolescent. Argenteuil Hospital Centre, Argenteuil, France b. ECSTRA Team, UMR-1153, Inserm, Paris Diderot University, Sorbonne Paris Cité
REVIEW RETURNED	29-Nov-2017

GENERAL COMMENTS	This is a very interesting metasynthesis about an important subject. I would like to congratulate the authors for their rigorous work regarding the methods they employed. if some results were expected (the need of a trustful relationship with the professional, the safe environment and so) some opened to new perspectives, especially the fourth analytical theme that crossed the perspectives of the stakeholders. discussion is very good with concrete outcomes in line with Thomas & Harden conception of Thematic synthesis Only some minor revisions are needed:  1. abstract: the first sentence (objectives) needs to be reformulate and in a more clearer way 2. introduction should be more developed, especially when the authors mentioned the negative impact on children's physical and mental health. some quantitative findings could be presented here. 3. Methods: as a qualitative researcher myself, I completely understand why authors choose to focus on qualitative evidence. However I think readers not familiar with it would need a definition of qualitative methods. 4. eligible criteria: you need to explain here why you excluded qualitative research from the "grey literature" and not only in the limitations. 5. results: results are not presented clearly enough, from a methodological perspective the transparency and trustworthiness of the results are perfect but, by choosing to focus on these aspects, the content of the results is less available, only with a brief summary of the 4 analytical themes but not a total description of each theme (except briefly in the Tables). According to me, presentation of the results should be redone in order to facilitate the reading.
--

REVIEWER	Ann-Katrin Meyrose University Medical Center Hamburg-Eppendorf Department of Child and Adolescent Psychiatry, Psychotherapy, and Psychosomatics
-----------------	---

	Child Public Health Germany
REVIEW RETURNED	14-Dec-2017

GENERAL COMMENTS	COMMENTS FOR AUTHORS Comments in general: This interesting manuscript employs the precursors as well as the acceptability of identification of children's exposure to intimate partner violence in healthcare and social care as well as the acceptability of initial responses to children's exposure to intimate partner violence. Data of 11 qualitative studies were appraised including descriptions of children's, parents' and healthcare/social care professionals' experiences. Recommendations for action and future research are discussed. This manuscript has many positive qualities including an important target group, comprehensive supplements and meaningful conclusions. However, the manuscript could be improved considerably if the following major and minor concerns are addressed. Comments per section: Abstract  • The abstract is partly not written in entire sentences. Thus, it is difficult to understand the main messages (refers to all sections despite the conclusion). Carefully worded and clear sentences would improve the comprehensibility. • Minor: In line 28/29, it is not clear to what "seven studies in the top tertiles" is related (to the Appraisal checklist?). Introduction  • It would be interesting to know whether there are special "risk groups" suffering from IPV (e.g. certain age groups, boys/girls, cultural differences) and whether the approaches to identification and initial responses are acceptable for all children in the same way. Methods  • Studies selection: How often did reviewer disagree? This is important to know, because only 10% were rated twice and discussed. If the agreement was low, the remaining 90% might be a problematic basis for further investigation. Results  • Critical appraisal: Supplement file 4 is mentioned as a reference that the quality of studies was good overall, but the basis for this decision is not clear. Supplement file 4 includes all criteria and the studies, which meet the criteria, but the underlying rationale is not described. Further, afterwards (line 22-28) only negative aspects are listed. • In general, it would be important to know which statements refer to which stakeholder. In many cases this becomes clear, but in some not (e.g. page 12, line 18 to 22; p. 12, line 38: recipients = children or mothers?). • In the introduction only the identification (1.) as well as the initial responses (2.) are mentioned as research questions, but the precursors seem to be an important part of the results. The manuscript would benefit from a clear and consistent structure in the introduction, results and discussion sections. • Page 13, first paragraph: All references are mentioned in the end. Do they all refer to all statements of the paragraph? If not, it would be good to reassign the references. • Page 13, line 21/22: "...between three studies and resolved...": Which three studies are meant? • Referring to the point of the introduction – are there
--

	differences between special groups (e.g. age groups, girls/boys) indicated by the included studies?  In general, less children's experiences were available compared to mothers' and health/social care providers' experiences. It would be valuable to know for which themes children could be included and for which not. This would be a clear statement that experiences were not available, so that nobody would conclude that children's experiences are not important. This aspect is discussed later, but it is also a result that information on children's opinion is lacking. Suggestions: 1. It would be good to include a short and clear statement, when children's opinion could not be included. 2. For an overview: The last column of table 2 could be divided into three columns – one for each stakeholder and the related studies. Minor: Table 2 is very interesting, but it should be mentioned in the text. Further, please check the numbering (e.g. 2.2). Discussion  Page 15, line 9: Here you talk about "parents", but in the results section mothers are mentioned mostly. The manuscript would benefit from a distinction between mothers, fathers (seem to be hardly considered), and parents. There might be important differences, which could be important for the interpretation and conclusion. Page 16, line 36/37: "In contrast, mothers and children were in favor of HCPs talking directly to children..." This statement does not fit to the results section (p.14, l.37) – "Children's preference to be asked ... which does not match any of the professional or parent viewpoints". "Varied ages and settings" (p.17, l. 41) are mentioned in the discussion, but this point is not discussed in the manuscript. This aspect is very interesting, more information would be valuable for the reader (compare comments above on the introduction and results sections). Can you talk about the generalizability of your results and the included stakeholder? Figures  There are two figures included in the manuscript (after the references). Both are not mentioned in the text, therefore, the relation between the figures and the text is not clear. In addition, please add a short description/title to every figure (and table). Despite these critical notes this manuscript is really interesting. A revision is necessary, but promising.
--	---

VERSION 1 – AUTHOR RESPONSE

Dear editors and reviewers,

Thank you for your email from 14 December 2018 enclosing the reviewers' comments. We would like to thank all reviewers for the positive feedback and constructive critique. We have carefully reviewed the comments and have revised the manuscript accordingly. Our point-by-point responses are given below. Changes to the manuscript are shown in the 'Marked copy' in red.

Tuija Helena Leppäkoski, Reviewer 1

1. Theoretical background is a little bit parsimonious. First, the concept and content of IPV (also called DV) was not clearly and accurately defined. Second, current research data supports the need to consider men and women as both potential victims and perpetrators when approaching IPV (e.g. Int J Public Health 2015;60:467-478; J Fam Viol 30:699-717). This study emphasize that the women are victims and the men are perpetrators. The choice of this point of view is necessary to justify.

We appreciate the Reviewer pointing out the need to include a clear definition of IPV. The WHO definition has now been included and the Introduction has been rewritten to emphasise that IPV is experienced by men and women:

Intimate partner violence (IPV) is a violation of human rights and widespread public health problem that is associated with impairment throughout the lifespan. It is defined as any behaviour by a current or former intimate partner associated with physical, sexual or psychological harm, including acts of physical aggression, sexual coercion, psychological abuse and controlling behaviours. Although IPV is experienced by both men and women, the morbidity and mortality related to IPV is highest among women

2. Second, violence against the child within the family is a serious and complex phenomenon and there are many factors that influence it. Further, it is often concealed. For this reason, I suggest that the authors will address the types and consequences of violence in a little more detail. (E.g. WHO; INSPIRE. (www.who.int/violence_injury_prevention/violence/inspire/en/) Further, IPV effects on the health and well-being of the whole family.

We appreciate this suggestion and have expanded the section referring to impairment for children exposed to IPV, within the constraints of the word limit:

Children's exposure to IPV is strongly associated with a broad range of emotional and behavioural problems, including internalising and externalising symptoms, as well as increased risk-taking behaviour and academic problems. Furthermore, such exposure among children can lead to physical health consequences, including injuries and death, when physical violence between caregivers directly involves children.

3. However, the structure of Methods section is somewhat confusing. Sometimes, it is difficult to understand and evaluate the points (paragraphs) the authors have made. It contains a lot of sub-headings and the text breaks down. Is it possible to combine them into larger entities? For example: search methods, selection criteria, data extraction and synthesis.

We have re-structured Methods section under three subheadings: Search methods, Studies selection, Analysis.

4. Timeframe in which the literature was selected is not mentioned in the text.

The timeframe of searches is reported in Table 1 ("Publication date inception to April 28th, 2016"). We have added the publications timeframe to Results (1st paragraph):

All papers were published between 2008 and 2015.

5. Moreover, there are many supplementary files, which must be read simultaneously with the text to understand what the authors have done. I think that supplementary file 3 (list of excluded papers and reasons for exclusion) is completely unnecessary and exclusion criteria can be said in the text.

"Studies reported in books ... were excluded as well as languages other than English, quantitative research and conference proceedings" (Page 5).

The Methods opening sentence explains why we did not include quantitative studies. Following the Reviewer's suggestion, we have removed supplementary file 3, amended PRISMA diagram, and explained in text (Studies selection, 1st paragraph) why we excluded non-English and non-peer-reviewed papers):

Exclusion of non-English papers and papers that have not been through the formal peer review system (e.g., books, conference papers, editorials, letters, general comment papers) was justified by limited resources and concerns about validity and reliability of non-peer-reviewed sources, respectively.

6. Supplementary file 5 (Characteristics of the 11 studies ...) there could be added two columns that briefly summarize the key findings of the each study and the M-CASP score.

Supplementary file 5 (now Supplementary file 2): we have added two columns into the table. Key findings are now summarised in column 'Primary themes relevant to the synthesis'. Column 'M-CASP score' shows numbers of Yes-No-Not sure answers. Total Yes M-CASP scores are shown in Figure 2 which is included in the main text.

7. Results: Things that appear in the first two sub-chapters (Searches & Critical appraisal) are perhaps better suited to the Methods section. For example, the flow chart of study selection process (Fig 1) as well as study quality assessment could be placed under Methods section.

We followed the CRD and PRISMA guidance and distinguished between the description of methods (reported in Methods) and reporting results of applying these methods (reported in Results): This systematic review follows the Centre for Reviews and Dissemination (CRD) and PRISMA guidance and adheres to the ENTREQ statement for reporting synthesis of qualitative research. To improve the presentation of results, we have re-structured this section under two subheadings: Study characteristics and methodological quality, Synthesis.

8. In the first chapter there is something contradictory between the text and source references (four papers are mentioned but three sources 32-34, lines 15-17) and between the text and supplementary file (Supplementary file 3 comprises list of excluded papers and reasons for exclusion).

We appreciate the Reviewer identifying this problem and have corrected the referencing to include all four papers. In response to the previous comment (n.5), we have removed Supplementary file 3 and amended PRISMA diagram accordingly.

9. The text that begins with the words "Two papers were from ..." can be placed well under Study characteristics.

Following the PRISMA guidelines, we have distinguished between studies and papers in which the studies were reported. Thus, the first paragraph of Results reports results of the searches and numbers of the identified papers. We have found that three studies were reported in multiple papers and explained the connection between studies and papers in this paragraph.

10. Discussion: What can be said about the generalizability and transferability of the research results? It is important to tell readers something about these.

We have added a paragraph to Discussion/Limitations about the reliability, objectivity and generalisability of the produced evidence (Strengths and limitations, 3rd paragraph):

The evidence we produced should be interpreted with caution, taking into consideration the following limitations. It is supported by only 11 studies from high- and middle-income countries, four of which had methodological shortcomings regarding reliability and nine were lacking objectivity. The selective nature of the stakeholders' sample makes our findings relevant to HCPs and children's social service providers, mothers who have experienced IPV and their children. All evidence supported by children's voices came from two studies.

11. On page 17 it is said that review has several limitations but only two (only papers published in English are included & only two of eleven studies reported children's voices) are mentioned. For example, the women survivors are over-represented in the review material. Further, "All relevant parents' quotations were from mothers", (page 9). This also might be a gap.

We appreciate the Reviewer drawing attention to this point and have added a paragraph to Discussion/Limitations about the selective nature of the stakeholders' sample (Strengths and limitations, 3rd paragraph):

The selective nature of the stakeholders' sample makes our findings relevant to HCPs and children's social service providers, mothers who have experienced IPV and their children.

12. Already known, that both men and women involved in IPV as both victims and perpetrators. Also mothers abuse their children and spouses, but this is not always revealed. Professional's obligation is to identify fathers' symptoms and signs of IPV as well, and provide services for them. In the future, it is important to extend research in this direction. This topic as a future research area, have not discussed.

The Reviewer has raised an important issue and we have now suggested including fathers who have experienced IPV in future research (Strengths and limitations, 4th paragraph):

Future studies should also recruit fathers who have experienced IPV.

13. Other remarks:

Page 24. The title of Figure 1 is missing. Likewise, there is no source reference. Where does this flowchart come from?

We have added the title to Figure 1.

Page 25. The figure number is missing.

We have added the figure number.

The reference to Table 2 is missing from the text page 9.

The reference to Figure 1 is in the first sentence of Results: "Sixteen papers reporting 11 studies were included (Figure 1)".

In Supplementary file 5, Characteristics of 11 studies (page 46), the reference number is obviously wrong. Szilassy (2015) is number [39] not [34].

The reference is corrected. Former online Supplementary file 5 is now Supplementary file 2.

14. What is new understanding and / or knowledge did this paper bring in compared to the previous ones concerning identification of and responding to children exposed to IPV, especially in international level? This was somewhat unclear and requires clarification.

We have summarized the new knowledge contributed by this review, and how it enhances guidance provided to health care professionals when identifying and responding to this issue in practice, using a current national Canadian project (VEGA) as an example:

Conclusion

The present analysis adds to the evidence base important, client-centred considerations drawn from qualitative research to enhance approaches to the identification of, and response to, children exposed to IPV, and their caregivers. Health and social care professionals should receive sufficient training and on-going individual and system-level support to recognize and respond safely to children exposed to IPV. Ideal identification and response approaches should be based on the WHO LIVES principles, integrated into a trauma- and violence-informed model of care. In Canada, findings from the present analysis have been integrated into professional guidance and curriculum in the Violence, Evidence, Guidance, Action (VEGA) Project.

John Devaney, Reviewer 2

15. p6 A 10% subset of the full text searches were checked - what level of agreement was there between the first and second reviewer?

We have reported the frequency of disagreements between two reviewers and highlighted that they were resolved through consensus (Studies selection, 2nd paragraph):

The first reviewer screened all full text papers, the second reviewers screened a 10% subset, and disagreements (27%) were resolved through discussion and consensus.

16. In some jurisdictions, such as the USA, child protection workers are not social workers. As such using the term 'social services' or 'social care practitioners' might appear misleading, as they are typically referred to as 'child protection services' and 'child protection workers'. This needs better explained in the opening sections as to how 'health practitioners' and 'social care practitioners' are defined.

To deal with differences in terminology across countries, we defined 'social service providers/professionals (SSPs)' as professional working in a broad range of services provided to advance adult or child welfare (including child protection/safeguarding services). To illustrate the range of such services, we have added a new table - Table 2. Summary characteristics of included studies (Results), and added a table note to clarify our definition (Studies selection, Table 1):

Note. Social services cover a range of services provided to advance adult and child welfare including child protection/safeguarding services.

We have also added sentence summarising the background of HCPs and SSPs ():

HCPs included physicians and nurses from primary and secondary health care. SSPs were drawn from children's social services, child protection services and unspecified settings.

17. Given that the paper deals with the issue of identification and initial response there is a need to at least acknowledge that some of the included studies took place in countries with mandatory laws for the reporting of child maltreatment, which might include children's exposure to domestic violence. This may influence whether professionals enquire about certain issues, and the required response. This needs discussed, as the context of the included studies may influence and moderate the findings.

We have added the following to Discussion:

While not the focus of this synthesis, it is important to note that mandatory reporting of child maltreatment laws may complicate or intertwine with strategies for inquiring about exposure to IPV, as children's exposure to IPV is a reportable exposure in some jurisdictions. We found that most HCPs were confused as to whether or not children's exposure to IPV was reportable in their jurisdiction. They felt anxiety about the reporting duties and thresholds for reporting children's exposure to IPV and requested training and resources on this topic. This finding is in line with the recent meta-synthesis about mandated reporters' experiences with reporting maltreatment. The authors offer recommendations for mitigating potential harms associated with the reporting processes. The strategies include disclosing reporting duties and the limits of confidentiality when providers start a relationship, consulting with child protection services in an anonymous manner when a provider is unsure if the suspected maltreatment is reportable, and – if the suspected maltreatment is reportable – discussing with the child/family how the provider will file a report (when it is safe for the child to do so) and likely child protection service responses to the report. Discussing reporting duties and limits of confidentiality should ideally occur before inquiry, to minimise feelings of betrayal that may emerge when a provider realises they must report. These recommended strategies are applicable to the reportable children's exposure to IPV.

18. Some of the papers (eg Stanley) are about the criminal justice response alongside the social care response - this is not clear in the presentation of findings or discussion.

The two Stanley's papers report on the study on police and children's social services' responses to children's exposure to IPV. For this synthesis, we extracted and re-analysed the primary data relevant to the social services' response to the police notifications. We have re-worded the description of the Stanley's study in Results (1st paragraph) and online Supplementary file 2 (Studies characteristics): Two papers drew on a study of police and children's social services responses to IPV incidents where children were present or resided in the household.

We have also added a sentence to Methods/Analysis clarifying how the primary data were selected against our inclusion criteria (Analysis, 2nd paragraph):

Where studies included varied participants, only data relevant to our inclusion criteria were considered.

Emiko Tajima, Reviewer 3

19. The authors have been thorough in addressing the requisite elements of a rigorous meta synthesis, including clear search terms, sources, and methods, clear criteria for inclusion, clear analytic approach and systematic assessment of study rigor. The synthesis is comprehensive and thematic findings are presented in an easily digestible manner, being shown in both table outline form and in more detailed, narrative form. This thoroughness has yielded a comprehensive, yet lengthy manuscript with some repetitiveness.

When addressing comments 3 and 7 from Reviewer 1, we have re-structured Methods and Results. We hope that the new structure is less repetitive.

20. The authors focused on two areas: "acceptable approaches" to identifying IPV exposure in children, and "initial responses" to identified IPV exposure in different settings, including clinical settings, social service departments, IPV shelters, and child welfare settings. They sought to examine experiences and perspectives of service providers, victims of IPV, and children. While it is commendable to examine experiences in a range of settings such as in the current synthesis, it also complicates the interpretation and application of the findings because the settings are quite different, and the context is therefore different. For example, the identification of IPV exposure in the context of a child protective services office is different from identification within a domestic violence shelter, and different still from identification in a doctor's office. The context differs, as do the potential consequences of identification. Professional service providers across those different settings have different perspectives, training, and understandings of IPV, and have very different roles in relation to the caregivers and their children. What works best and is perceived as most appropriate in one setting may not be regarded as appropriate or desirable in another setting. It is also not clear the level of "crisis" that the respondents might have been in at the time they participated in the studies. This

aspect of the context matters as well. To their credit, the authors made an effort to examine cultural context factors.

When addressing comment 10 from Reviewer 1, we have added a paragraph to Discussion/Limitations about the generalisability of the produced evidence.

21. Even greater attention to race and culture in such a synthesis would be a welcome addition to the literature.

We have extended the Discussion to include the finding on the need for multi-language resources for patients/clients.

Enabling conditions included sufficient training and multiple language-versions materials and embedding the work of identifying and responding to children's exposure to IPV into the context of under-resourced services.

22. It is not clear to me whether the practitioners who participated in the studies were all mandated reporters of child abuse – it complicates the synthesis that it's unclear whether the precursors identified apply to mandated reporters of child abuse or to any practitioner in the studied settings. It is notable that practitioners reported considerable confusion themselves about whether or not they needed to report identified child abuse, and lack of clarity about how to define reportable child abuse, including whether or not IPV exposure itself was considered child abuse. I would not define IPV exposure as child abuse per se, however it is clear from the quotations that at least one of the practitioners defined it as such. This synthesis offers critical findings about the extent of confusion among practitioners about reporting requirements and documentation recommendations – this is something that administrators and policy makers should address.

The authors of the included studies did not report the reporting status of the participants. In three studies, mothers and professionals talked about uncertainty and confusion regarding reporting duties and thresholds for children's exposure to IPV; professionals requested relevant training and resources. These views are captured by the descriptive theme Experiences of initial response to children's exposure to IPV/Reporting (inline Supplementary file 4), interim analytical theme Gaps in knowledge and skills/Mandatory reporting (online Supplementary file 5) and final analytical theme Precursors for acceptable identification and response/Professional training and resources (Table 3). The analytical theme is summarised in the Synthesis/Analytical themes, 4th paragraph.

When addressing comment 17 from Reviewer 2, we have discussed the potential impact of the mandatory reporting on the approaches to identification and response to children's exposure to IPV.

23. It is a strength of the synthesis that the authors sought to explore the perspectives of multiple stakeholders, namely health and social service providers, caregivers who experienced IPV, and their children. It is useful to better understand patient /client and constituent perspectives and use this knowledge to inform clinical practice and policy. It is particularly valuable to see the points of alignment and also the areas of disagreement, especially comparing professionals and clients. Although one would hope that hospital and social service administrators would be interested in patient/client perspectives and preferences, they may be more responsive to data on positive outcomes than to data on patient/client beliefs and preferences. To change policy and practice, it may be necessary to also synthesize findings on factors that increase positive outcomes that administrators and policy-makers may care most about (e.g., positive patient health outcomes, reduced IPV, greater awareness of IPV effects on children, fewer hospital visits etc.).

We agree with the reviewer on this point, however to date these efforts have been thwarted by a evaluative studies which often measure narrow outcomes that do not fully reflect the priorities of those commissioning, delivering and using services.[1] It is our experience that there is a good deal of overlap in concepts of success held service users and service commissioners, however service commissioners often highlight the importance of value for money, which can only be adequately assessed using quantitative research methods.

Jordan Sieboni, Reviewer 4

24. Only some minor revisions are needed: abstract: the first sentence (objectives) needs to be reformulate and in a more clearer way.

We have re-worded the first sentence of the abstract:

To synthesise evidence on the acceptable identification and initial response to children's exposure to intimate partner violence (IPV) from the perspectives of providers and recipients of healthcare and social services.

25. introduction should be more developed, especially when the authors mentioned the negative impact on children's physical and mental health. some quantitative findings could be presented here . We have re-written the Introduction when responding to comments 1 and 2 from Reviewer 1.

26. Methods: as a qualitative researcher myself, I completely understand why authors choose to focus on qualitative evidence. However I think readers not familiar with it would need a definition of qualitative methods.

We have added a definition of qualitative methods with supporting reference:

Qualitative research explores peoples' own experiences and perspectives through analysing textual or visual material obtained while talking to people or observing them.

27. eligible criteria: you need to explain here why you excluded qualitative research from the "grey literature" and not only in the limitations.

We have explained in Methods why we excluded non-peer-reviewed papers (Studies selection, 1st paragraph):

Exclusion of non-English papers and papers that have not been through the formal peer review system (e.g., books, conference papers, editorials, letters, general comment papers) was justified by limited resources and concerns about validity and reliability of non-peer-reviewed sources, respectively.

28. results: results are not presented clearly enough, from a methodological perspective the transparency and trustworthiness of the results are perfect but, by choosing to focus on these aspects, the content of the results is less available, only with a brief summary of the 4 analytical themes but not a total description of each theme (except briefly in the Tables). According to me, presentation of the results should be redone in order to facilitate the reading.

We have re-formatted the Results/Synthesis section to describe each final analytical theme before Table 3. Unfortunately, the words limit does not allow us sufficient space to define all descriptive themes and interim analytical themes in the main text. They are summarised in online Supplementary files 4 and 5.

Ann-Katrin Meyrose, Reviewer 5

29. Abstract. The abstract is partly not written in entire sentences. Thus, it is difficult to understand the main messages (refers to all sections despite the conclusion). Carefully worded and clear sentences would improve the comprehensibility.

We have re-written the abstract in sentences.

30. Abstract. Minor: In line 28/29, it is not clear to what "seven studies in the top tertiles" is related (to the Appraisal checklist?).

We have removed this result.

31. Introduction. It would be interesting to know whether there are special "risk groups" suffering from IPV (e.g. certain age groups, boys/girls, cultural differences) and whether the approaches to identification and initial responses are acceptable for all children in the same way.

When addressing comments 1 and 2 from Reviewer 1, we have re-written the Introduction.

Unfortunately, the word limit does not allow us to elaborate on risk factors for children's exposure to IPV. Instead, we provide reference to a more specific overview about the epidemiology of children's exposure to IPV, including prevalence, risk and protective factors, and associated impairment, as well as strategies for identification, and interventions for prevention of exposure and impairment.[2]

32. Methods. Studies selection: How often did reviewer disagree? This is important to know, because only 10% were rated twice and discussed. If the agreement was low, the remaining 90% might be a problematic basis for further investigation.

When addressing comment 15 from Reviewer 2, We have reported the frequency of disagreements between two reviewers and highlighted that they were resolved through consensus (Studies selection, 2nd paragraph).

33. Results. Critical appraisal: Supplement file 4 is mentioned as a reference that the quality of studies was good overall, but the basis for this decision is not clear. Supplement file 4 includes all criteria and the studies, which meet the criteria, but the underlying rationale is not described. We have explained that our conclusion of the overall quality of the primary data is based on the total M-CASP score being $\geq 15/20$ (Study characteristics and methodological quality, 3rd paragraph): Of 11 studies, seven scored ≥ 15 out of 20 on the M-CASP indicating their overall good methodological quality (Figure 2, online Supplementary file 3). The main shortcomings identified were in the M-CASP domains of reliability and objectivity. Thus, the authors did not justify their choices of study design, research methods, participant selection and recruitment. Only two studies described strategies for establishing neutrality.

34. Results. In general, it would be important to know which statements refer to which stakeholder. In many cases this becomes clear, but in some not (e.g. page 12, line 18 to 22; p. 12, line 38: recipients = children or mothers?).

We have clarified which stakeholder group contributed to each analytical theme throughout the Results/Synthesis.

35. Results. In the introduction only the identification (1.) as well as the initial responses (2.) are mentioned as research questions, but the precursors seem to be an important part of the results. The four analytical themes answer the two research questions (What approaches to identification of children's exposure to IPV are acceptable to children, non-abusing parents and professionals? And What initial responses to children identified as being exposed to IPV are acceptable to children, non-abusing parents and professionals?). The precursors theme is relevant to both research questions through illuminating factor and conditions enabling an acceptable identification/initial response.

36. The manuscript would benefit from a clear and consistent structure in the introduction, results and discussion sections.

Following advice from other Reviewers (comments 3, 7), we have re-structured the Introduction, Results and Discussion.

37. Page 13, first paragraph: All references are mentioned in the end. Do they all refer to all statements of the paragraph? If not, it would be good to reassign the references.

We have re-assigned the references.

38. Page 13, line 21/22: "...between three studies and resolved...": Which three studies are meant?

We have added references to the relevant studies.

39. Referring to the point of the introduction – are there differences between special groups (e.g. age groups, girls/boys) indicated by the included studies?

The primary studies did not report analysis by age groups or boys/girls. When extracting the raw data, it was not always possible to establish the age of the participant; a name could be an indicator of the child's sex. We did not observe differences between boys and girls when synthesising the primary data.

40. In general, less children's experiences were available compared to mothers' and health/social care providers' experiences. It would be valuable to know for which themes children could be included and for which not. This would be a clear statement that experiences were not available, so that nobody would conclude that children's experiences are not important. This aspect is discussed later, but it is also a result that information on children's opinion is lacking. Suggestions: 1. It would be good to include a short and clear statement, when children's opinion could not be included.

Table 3 (former table 2) shows which stakeholder groups contributed to each theme and subtheme.

We have re-written a summary sentence explaining which themes were not supported by children's voices (Analytical themes, 2nd paragraph):

Not all participant groups contributed equally to the final analytical themes. Thus, professionals' perspectives were presented across 15/17 subthemes, while mothers informed 13/17 and children 9/17. Children's quotes were not available for most subthemes covering an acceptable initial response.

41. Suggestions: 2. For an overview: The last column of table 2 could be divided into three columns – one for each stakeholder and the related studies.

We have divided the last column of Table 2 (now Table 3) into three to show which stakeholder group contributed to each final analytical theme.

42. Minor: Table 2 is very interesting, but it should be mentioned in the text. Further, please check the numbering (e.g. 2.2).

We have included reference to Table 2 (now Table 3) in the text and corrected the numbering.

43. Discussion. Page 15, line 9: Here you talk about “parents”, but in the results section mothers are mentioned mostly. The manuscript would benefit from a distinction between mothers, fathers (seem to be hardly considered), and parents. There might be important differences, which could be important for the interpretation and conclusion.

We have distinguished between parents/mothers/fathers throughout the manuscript. From the 5th paragraph of Results where we report that all relevant parents’ quotations came from mothers, we replaced ‘parents’ with ‘mothers’. We also amended the title of the manuscript to reflect this important finding.

44. Discussion. Page 16, line 36/37: “In contrast, mothers and children were in favor of HCPs talking directly to children...” This statement does not fit to the results section (p.14, l.37) – “Children’s preference to be asked ... which does not match any of the professional or parent viewpoints”. The first quote is about talking directly to children at consultation, while the second one is about risk assessment – asking children whether they feel safe at all environments they live. We have re-worded the risk assessment sentence in Results (Analytical themes, 11th paragraph):

Children’s preferences for the assessment of risk in all environments are represented by one boy’s voice which does not match any of the professional or parent viewpoints.

45. Discussion. “Varied ages and settings” (p.17, l. 41) are mentioned in the discussion, but this point is not discussed in the manuscript. This aspect is very interesting, more information would be valuable for the reader (compare comments above on the introduction and results sections).

We have reported children’s ages in Results and added information about settings which they were recruited from (Studies characteristics and methodological quality, 2nd paragraph):

Overall, the studies involved 42 children and young people aged 8 to 24 (19 from IPV and social services, 23 from general practice), 220 parents (212 mothers), and 251 professionals (113 health care, 42 social services, and 96 mixed samples).

Table with studies characteristics (online Supplementary file 4) shows these data in detail.

46. Can you talk about the generalizability of your results and the included stakeholder?

We have addressed this issue when responding to comment 10 from Reviewer 1. The generalisability of the produced evidence is limited by the small number of included studies, their methodological limitations and underrepresentation of fathers and children (Strengths and limitations, 3rd paragraph): The evidence we produced should be interpreted with caution, taking into consideration the following limitations. It is supported by only 11 studies from high- and middle-income countries, four of which had methodological shortcomings regarding reliability and nine were lacking objectivity. The selective nature of the stakeholders’ sample makes our findings relevant to HCPs and children’s social service providers, mothers who have experienced IPV and their children. All evidence supported by children’s voices came from two studies.

47. There are two figures included in the manuscript (after the references). Both are not mentioned in the text, therefore, the relation between the figures and the text is not clear.

We have added references to Figures 1 and 2 into the main text.

48. In addition, please add a short description/title to every figure (and table).

We have added titles and legends to Figures and tables.

We would like to express our great appreciation to you and the Reviewers for the comments on our manuscript. If you have any further queries, please do not hesitate to contact us.

Kind regards,
Natalia Lewis

VERSION 2 – REVIEW

REVIEWER	Tuija Helena Leppäkoski, PhD University of Tampere, Nursing Science / Health Sciences, Faculty of Social Sciences, Finland.
REVIEW RETURNED	25-Jan-2018
GENERAL COMMENTS	The text has been clarified and improved throughout. I have no further remarks. I would be willing to accept it.
REVIEWER	Ann-Katrin Meyrose University Medical Center Hamburg-Eppendorf
REVIEW RETURNED	25-Jan-2018
GENERAL COMMENTS	This manuscript was improved considerably by addressing the reviewers' recommendations. Especially the focus of the manuscript is more precise and editing the tables/figures increased the comprehensibility.
REVIEWER	Emiko Tajima, Associate Professor University of Washington, School of Social Work USA
REVIEW RETURNED	02-Feb-2018
GENERAL COMMENTS	I believe that the revised manuscript addresses the major concerns I raised, at least those that the authors were able to address. Any reviewer concerns that the authors could not address (e.g., if original studies did not specify certain information) should be noted as limitations of the paper.

VERSION 2 – AUTHOR RESPONSE

Dear Assistant Editor and Reviewers

Thank you for your email from 28 February 2018 enclosing the reviewers' comments on the revised manuscript. Our response to Reviewer 3 is given below.

Reviewer: 3

Any reviewer concerns that the authors could not address (e.g., if original studies did not specify certain information) should be noted as limitations of the paper.

We have added limitation about unavailable data on mandated reporters in primary studies:

Some important information (e.g., mandatory reporting status of professionals) was not specified.